# LAMDA: Unified Language-Driven Multi-Task Domain Adaption

## Abstract

Unsupervised domain adaption (UDA), as a form of transfer learning, seeks to adapt a well-trained model from supervised source domains to an unlabeled target domain. However, most existing UDA approaches have two limitations. Firstly, these approaches assume that the source and target domains share the same language vocabulary, which is not practical in real-world applications where the target domain may have distinct vocabularies. Secondly, existing UDA methods for core vision tasks, such as detection and segmentation, differ significantly in their network architectures and adaption granularities. This leads to redundant research efforts in developing specialized architectures for each UDA task, without the ability to generalize across tasks. To address these limitations, we propose the formulation of unified language-driven multi-task domain adaption (LAMDA). LAMDA incorporates a pre-trained vision-language model into the source domains, allowing for transfer to various tasks in the unlabeled target domain with different vocabularies. This eliminates the need for multiple vocabulary-specific vision models and their respective source datasets. Additionally, LAMDA enables unsupervised transfer to novel domains with custom vocabularies. Extensive experiments on various segmentation and detection datasets validate the effectiveness, extensibility, and practicality of the proposed LAMDA.

## 1 Introduction

In recent years, there has been remarkable progress in achieving high performance in diverse visual tasks, thanks to the utilization of large-scale fine-grained annotated datasets. Nevertheless, the process of collecting and annotating these extensive training data is not only financially burdensome but also demanding in terms of time and effort. To overcome this challenge, the research community has turned its attention towards unsupervised domain adaption (UDA) techniques, which aim to adapt a vision model that has been pre-trained on labeled source domains to target domains utilizing unlabeled target images (Hoyer et al. (2023); Zhang et al. (2023); He et al. (2021)).

Notwithstanding the significant advancements made in the field of unsupervised domain adaption (UDA), there remain two prominent limitations within the existing literature. Firstly, a prevalent assumption in most UDA tasks is that the source and target domains share a common vocabulary. However, this assumption becomes a significant challenge when dealing with target domains that possess distinct vocabularies, severely constraining the flexibility and efficiency of unsupervised transfer. Secondly, it is worth noting that existing UDA methods for core vision tasks, such as detection and segmentation, exhibit notable disparities in terms of network architectures and adaption granularities, as evidenced in previous studies (Hoyer et al. (2023); Huang et al. (2023)). Although these specialized architectures have propelled the progress of individual tasks, they lack the versatility to generalize across different tasks. Consequently, redundant research efforts, along with hardware optimization, are expended on developing specific architectures for each UDA task, without the existence of a mature unified UDA framework that encompasses both tasks.

In light of these limitations, for the first time, we propose the novel formulation of unified language-driven multi-task domain adaptation (LAMDA), inspired by vision-language models (VLMs) (Radford et al. (2021b)) which have demonstrated their efficacy in enabling open-vocabulary visual recognition through the fusion of image and text reasoning. Building upon this, LAMDA serves as an unsupervised domain adaptation (UDA) framework that harnesses the power of a pre-trained

VLM in the source domains, facilitating its transfer to diverse unlabeled target domains. Notably, LAMDA stands out by requiring just a single pre-trained VLM to adapt to target domains characterized by varying vocabularies. This eliminates the need for multiple vocabulary-specific vision models, along with their associated source datasets. Moreover, LAMDA offers the unique advantage of unsupervised transfer to novel domains featuring custom vocabularies. By alleviating the burdensome requirements of extensive image annotation, LAMDA enhances deep network training capabilities, enabling effective handling of a wide range of visual recognition tasks. Importantly, LAMDA serves as a unified UDA framework for both detection and segmentation, surpassing the limitations of specialized architectures that are task-specific. It outperforms these specialized architectures across various UDA tasks, while maintaining ease of training and adaptability to every UDA task.

Overall, our contributions are summarized as follows: First and foremost, to the best of our knowledge, LAMDA represents the pioneering language-driven multi-task framework for unified segmentation and detection domain adaption. Secondly, we introduce the hierarchical visual-language alignment (HVA), to enhance the language-driven learning by leveraging both intra-source domain and inter-source-target domain information, along with promptable language-task learning (PLL) to mitigate the inter-task differences and inter-domain discrepancies in context distributions. Finaly, comprehensive experiments are conducted to demonstrate the universal effectiveness and practicality of LAMDA in the domains of both segmentation and detection tasks.

## 2 RELATED WORK

### 2.1 UNSUPERVISED DOMAIN ADAPTION

Unsupervised domain adaption (UDA) is a form of transfer learning that seeks to adapt a model, previously trained on labeled source domains, to unlabeled target domains (Bengio (2012)). Given the prevalence of domain gaps across various vision applications, UDA methods have found wide-ranging applications in major vision tasks, such as semantic segmentation (Hoyer et al. (2022b)), object detection (Chen et al. (2021); Li et al. (2022b)), and image classification (Hoyer et al. (2022a)), and have even been extended to other tasks like instance segmentation (Deng et al. (2022)). However, existing UDA methods primarily concentrate on designing specialized networks for single-task learning due to the differences in output formats and network architectures across different tasks. Consequently, there is a lack of flexibility in generalizing these methods to other tasks.

### 2.2 LANGUAGE-DRIVEN TASK

The application of language-driven techniques to existing tasks represents a dynamic and constantly evolving research domain that continues to push boundaries. Notable tasks in this field include, but are not limited to, segmentation (Li et al. (2022a)), image editing (Fu et al. (2022)), and style transfer (Fu et al. (2022)). The emergence of CLIP (Radford et al. (2021a)) has demonstrated the potential of language assistance in traditional vision tasks that traditionally have no explicit connection to language. However, the application of language-driven techniques to domain adaption remains relatively unexplored. Only one previous work (Huang et al. (2023)) has introduced the concept of open-vocabulary domain adaption, but their applicability is limited to image recognition tasks alone. To the best of our knowledge, LAMDA represents the pioneering comprehensive framework that formulates a unified approach, integrating various language-driven settings, to multiple core vision UDA tasks.

## 3 METHOD

### 3.1 PRELIMINARY OF UDA

In classical single-source UDA, a model $f_\theta$ can be trained on source domain $\mathbb{S}$ while adapted to the unlabeled target domain $\mathbb{T}$. In the source domain $\mathbb{S} = \{(x_{k^\mathbb{S}}^\mathbb{S}, y_{k^\mathbb{S}}^\mathbb{S})\}(k^\mathbb{S} \in [1, N^\mathbb{S}])$, where $x_{k^\mathbb{S}}^\mathbb{S}$, $y_{k^\mathbb{S}}^\mathbb{S}$ and $N^\mathbb{S}$ represent the $k^\mathbb{S}$-th image, its label and number of images in the source domain, respectively. The unlabeled target domain $\mathbb{T} = \{x_{k^\mathbb{T}}^\mathbb{T}\}(k^\mathbb{T} \in [1, N^\mathbb{T}])$ is similar. UDA is applicable to various

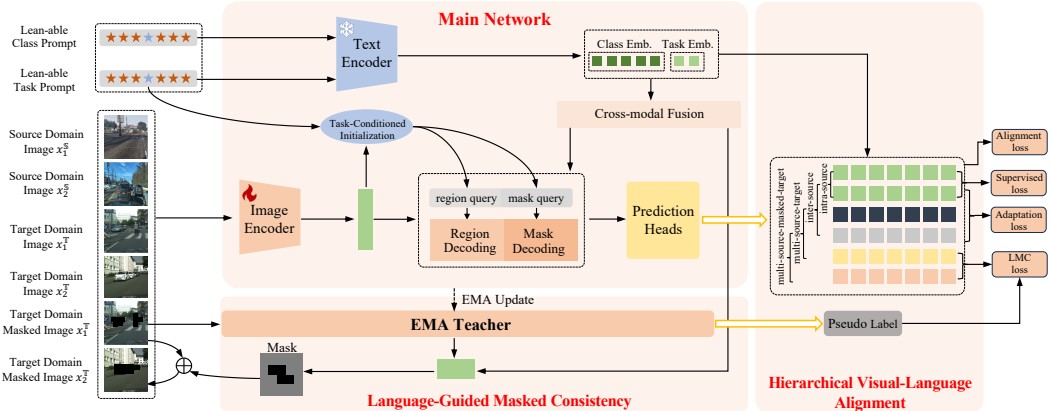

Figure 1: The overview of the proposed LAMDA, which can be semantically divided into three parts: Main Network, Language-Guided Masked Consistency (LMC), and Hierarchical Visual-Language Alignment (HVA).

vision tasks, for semantic segmentation, the overall training loss is typically as,

$$\mathcal{L} = \mathcal{L}^{\mathbb{S}} + \lambda\mathcal{L}^{\mathbb{T}} = \sum_{k^{\mathbb{S}}=1}^{N^{\mathbb{S}}} \frac{1}{N^{\mathbb{S}}} \mathcal{L}^{ce}(f_\theta(x_{k^{\mathbb{S}}}^{\mathbb{S}}), y_{k^{\mathbb{S}}}^{\mathbb{S}}) + \lambda^{\mathbb{T}} \sum_{k^{\mathbb{T}}=1}^{N^{\mathbb{T}}} \frac{1}{N^{\mathbb{T}}} \mathcal{L}_{k^{\mathbb{T}}}^{\mathbb{T}} \tag{1}$$

where $\mathcal{L}^{\mathbb{S}}, \mathcal{L}^{\mathbb{T}}$ are the supervised source loss and unsupervised adaption loss. $\lambda^{\mathbb{T}}$ is loss weight. $\mathcal{L}^{ce}$ is the basic cross-entropy loss, and can be replaced with box regression and classification loss for object detection. $\mathcal{L}_{k^{\mathbb{T}}}^{\mathbb{T}}$ is defined according the UDA strategy (Hoyer et al. (2023)) such as adversarial training or self-training. In the classical UDA, both the source and target domains share the common vocabulary.

## 3.2 Unified Language-Driven Multi-Task UDA

Our work focus on building up the novel unified language-driven multi-task UDA (LAMDA) framework for both semantic segmentation and object detection. Let $\mathbb{V}$ be the language category set in source domain, and the goal of LAMDA is to learn and adapt to perform segmentation/detection on visual concepts in $\mathbb{V}$ in training, and able to generalize to unseen categories on the unlabeled target domain.

As depicted in Figure 1, the proposed framework can be conceptually divided into three components: Main Network, Language-Guided Masked Consistency (LMC), and Hierarchical Visual-Language Alignment (HVA). The Main Network is responsible to receive the images from both source and target domains for supervised source training, unsupervised domain adaption respectively, where we formulate a unified proposal decoder for both semantic segmentation and object detection. Since there is no ground truth supervision for the target domain adaption, inspired by the success of masked image modeling (MIM) in self-supervised learning, we propose to specifically enhance the exploration of context relations on the target domain with LMC to provide additional clues for robust self-representation with similar local appearance. Finally, HVA is proposed to enable target domain with open-vocabulary understanding, from the visual-language alignment learning.

### 3.2.1 Main Network

The Main Network contains the following three parts. We aim to construct a unified framework for both semantic segmentation and object detection domain adaption. Note that the framework is able to easily extend other core vision tasks such as instance segmentation and depth estimation.

**Text Encoder:** Given the language category set $\mathbb{V}$, the text encoder represents each of its text item into text embeddings. Generally, the text encoder can be usual language-based architectures, here we follow the common practice and exploit the pretrained Contrastive Language–Image Pre-training

(CLIP) (Radford et al. (2021a)). According to characteristic of CLIP, the order of input texts is irrelevant to the set of output vectors.

**Image Encoder**: As a common block, image encoder takes image as input and produces pixel-wise visual embeddings, generally, with downsampling for the sake of memory efficiency. Formally, given an input image $\mathbb{R}^{H_0 \times W_0 \times 3}$, the output visual embedding can be denoted as $\mathbb{R}^{H \times W \times C}$, where $H = \frac{H_0}{d}, W = \frac{W_0}{d}$, $d$ is the downsampling rate. We exploit the Swin (Liu et al. (2021)) transformer as image encoder in our framework.

**Unified Proposal Decoder**: In order to unify the training of both semantic segmentation and object detection, we formulate the unified proposal decoder (UPD), which queries the visual features extracted by the image encoder into visual concepts and class-agnostic proposals for both the tasks. Due to the intrinsic discrepancies between the segmentation and detection: the former requires the recognition of both foreground and background while the latter focuses solely on localizing foreground objects, we exploit two individual queries to avoid the task conflicts that may significantly degrade performance. As shown in Figure 1, UPG is a unified decoder which consists of both the region decoding (for detection) and mask decoding (for segmentation) capability, corresponding to individual region query and mask query.

**Promptable Language-Task Learning**: Due to the inter-task differences between segmentation and detection, it is very hard for a single architecture to take into account both the language-driven UDA tasks in the absence of concrete task guidance, despite with UPD. Thus, we further adopt the Promptable Language-Task Learning (PLL) mechanism to collaboratively embed the language and task prompts. By this way, the multiple tasks are able to be encapsulated in a unified framework which effectively disentangles the parameter spaces to avoid the training conflicts in different tasks. Meanwhile, PLL is able to realize dynamic language vocabulary to popularize the framework to generalize to broader unseen categories and improve open-domain performance. In concrete, we formulate the prompt template as $P(\star) = [*, *, ..., \star, ..., *, *]$, where $\star$ is task text (e.g. semantic segmentation, object detection) or category text, and $*$ is learn-able vector. Then we model a joint language-task textual space with the general pre-trained CLIP Text Encoder $\mathbf{T}$, to get the multi-granularity embeddings $E$:

$$E = Cat(E^T, E^C) = Cat(\mathbf{T}(P(T)), \mathbf{T}(P(C))) \tag{2}$$

where $Cat(\cdot)$ indicates the concatenation operation. $E^T, E^C$ are the language category and task text embedding respectively. It is worth noting that the input category can be arbitrary, so $E$ can seamlessly adapt to unseen categories for open vocabulary segmentation and detection.

### 3.2.2 Language-Guided Masked Consistency

Predicting withheld tokens of a masked input sequence/image has shown to be a powerful self-supervised pretraining task in both natural language processing and computer vision. Inspired by the success of MLM and MIM, we propose the Language-Guided Masked Consistency (LMC) to provide additional clues for robust recognition of classes with similar local appearances from different parts of the image. This can be local information, which originates from the same image patch as the corresponding cell in the feature map, or context information, which comes from surrounding image patches that can belong to different parts of the object or its environment.

In order to build the LMC self-learning paradigm, firstly, we utilize a exponential moving average (EMA) teacher $f'_{\theta}$ with smoothing factor $\alpha$ on the weight of $f_{\theta}$:

$$f'_{\theta;t+1} \leftarrow \alpha f'_{\theta;t} + (1 - \alpha) f_{\theta;t} \tag{3}$$

where $t$ indicates the training step. By this way, EMA teacher is able to obtain the enhanced context learning capability from $f_{\theta}$, and exploit both the context and the intact local appearance information to generate pseudo labels of higher quality.

Different from the previous MIM using randomly sampling which may not provide the important clues needed to reconstruct the foreground objects (in the case of remaining visible patches only containing backgrounds), LMC introduces the language-guided masking paradigm to improve the reconstruction efficiency, by making more rational decisions on which patches to mask. Specifically, we query the visual concept $F \in \mathbb{R}^{S \times d}$ ($S$ is the sequence length and $d$ is the embedding dimension)

from the last layer of image encoder of $f_\theta^{'}$ with the language embedding $E^C \in \mathbb{R}$, to discriminate these semantically important patch clues:

$$M = \frac{1}{H} \sum_j^H softmax(\frac{(E_j^C W_{Q;j})([E_j^C, F]W_{K;j})^\top}{\sqrt{d}}) \tag{4}$$

where $H$ is the number of self-attention heads, $W_{Q;j}, W_{K;j}$ are the projection matrices of query and key respectively. $[\cdot, \cdot]$ is the concatenation operation, for simplicity we omit the MLP layers to unify the feature dimensions. $M$ is the language-visual cross-modal attention map and the magnitude of its elements is able to reflect the weights of the corresponding image patch contributing to the output feature of image encoder. Thus we utilize $M$ to guide the masking location on image patches with a multi-normal distribution sampling, and finally obtain the masked target image $\hat{x}^\mathbb{T}$ by masking the target domain image $x^\mathbb{T}$.

In order to maintain the self-consistency of the masked target image using the remaining context clues, we formulate the LMC loss as:

$$\mathcal{L}^{lmc} = \lambda^{lmc}\mathcal{L}^{ce}(f_\theta(\hat{x}^\mathbb{T}), f_\theta^{'}(x^\mathbb{T})) \tag{5}$$

where $f_\theta^{'}(x^\mathbb{T})$ is the pseudo label produced by the teacher network on the complete target domain image $x^\mathbb{T}$. Considering that the pseudo label may not be reliable especially in the very beginning of training, we add the quality weight $\lambda^{lmc}$ which defined as the maximum softmax probability in $f_\theta^{'}$ prediction.

### 3.2.3 Hierarchical Visual-Language Alignment

In order to take full advantage of both source domain and target domain to enhance the language-driven open-vocabulary learning while mitigating the inter-domain discrepancies in context distributions, we propose the Hierarchical Visual-Language Alignment (HVA) to align all the proposals (detection/segmentation) with language vocabulary, then leverage CLIP to perform zero-shot classification on the proposals. In each training batch, the input images reflect three levels of domain hierarchy: intra-source, inter-source. With the language-task prompt guidance, we obtain all the generated proposals by UPD, which come from both source and target domains.

Formally, given the input images set $\mathbb{I} = \{x_1^\mathbb{S}, x_2^\mathbb{S}, x_1^\mathbb{T}, x_2^\mathbb{T}, \hat{x}_1^\mathbb{T}, \hat{x}_2^\mathbb{T}\}$, where $x_1^\mathbb{S}, x_2^\mathbb{S}$ are two individual images in source domain, $x_1^\mathbb{T}, x_2^\mathbb{T}$ are two individual images in target domain respectively, $\hat{x}_1^\mathbb{T}, \hat{x}_2^\mathbb{T}$ are masked $x_1^\mathbb{T}, x_2^\mathbb{T}$ by LMC. Except for $\hat{x}_1^\mathbb{T}, \hat{x}_2^\mathbb{T}$ which are only used for LMC, we obtain all the proposal outputs

$$O = \{o_{1;1}^\mathbb{S}, o_{1;2}^\mathbb{S}, ..., o_{1;p_1^\mathbb{S}}^\mathbb{S}, o_{2;1}^\mathbb{S}, o_{2;2}^\mathbb{S}, ..., o_{2;p_2^\mathbb{S}}^\mathbb{S}, o_{1;1}^\mathbb{T}, o_{1;2}^\mathbb{T}, ..., o_{1;p_1^\mathbb{T}}^\mathbb{T}, o_{2;1}^\mathbb{T}, o_{2;2}^\mathbb{T}, ..., o_{2;p_2^\mathbb{T}}^\mathbb{T}\} \tag{6}$$

which corresponds to the outputs of input images respectively. $p_1^\mathbb{S}, p_2^\mathbb{S}, p_1^\mathbb{T}, p_2^\mathbb{T}$ are the number of proposals in the corresponding outputs respectively. Finally, we set up the cross-modal contrast-alignment to enforce the alignment between embedded visual proposal representations $O$ and the language-task text embedding $E$, which ensures that visual embeddings and its corresponding text tokens are closer in the feature space compared to embeddings of unrelated tokens. Formally the contrastive loss is defined as:

$$\mathcal{L}^{align} = \mathcal{L}_O^{align} + \mathcal{L}_E^{align} = - \sum_m^{[\mathbb{S},\mathbb{T}]} \sum_{i=1}^{p^m} \frac{1}{L} \sum_{j=1}^{L} log \left( \frac{exp(o_i^{m\top}t_j/\tau) \times 1(y_i == j)}{\sum_{k=1}^{L} exp(o_i^{m\top}t_k/\tau)} \right)$$

$$- \sum_m^{[\mathbb{S},\mathbb{T}]} \sum_{i=1}^{L} \frac{1}{p^m} \sum_{j=1}^{p^m} log \left( \frac{exp(t_i^\top o_j^m/\tau) \times 1(y_j == i)}{\sum_{k=1}^{p^m} exp(t_i^\top o_k^m/\tau)} \right) \tag{7}$$

$$y_i = \arg\max_j o_i^{m\top}t_j, \quad y_j = \arg\max_i t_i^\top o_j^m$$

where $\tau$ is the temperature parameter.

### 3.2.4 Overall Loss Formulation

Therefore, the overall loss formulation include four parts: segmentation/detection supervised loss $\mathcal{L}^\mathbb{S}$ for source domain, basic unsupervised adaption loss $\mathcal{L}^\mathbb{T}$, LMC loss $\mathcal{L}^{lmc}$ and visual-language

| Method | Road | S.walk | Build. | Wall | Fence | Pole | Tr.Light | Sign | Veget. | Terrain | Sky | Person | Rider | Car | Truck | Bus | Train | M.bike | Bike | mIoU |
|---|---|---|---|---|---|---|---|---|---|---|---|---|---|---|---|---|---|---|---|---|
| **Synthetic-to-Real: GTA→Cityscapes (Val.)** | | | | | | | | | | | | | | | | | | | | |
| ProDA | 87.8 | 56.0 | 79.7 | 46.3 | 44.8 | 45.6 | 53.5 | 53.5 | 88.6 | 45.2 | 82.1 | 70.7 | 39.2 | 88.8 | 45.5 | 59.4 | 1.0 | 48.9 | 56.4 | 57.5 |
| DAFormer | 95.7 | 70.2 | 89.4 | 53.5 | 48.1 | 49.6 | 55.8 | 59.4 | 89.9 | 47.9 | 92.5 | 72.2 | 44.7 | 92.3 | 74.5 | 78.2 | 65.1 | 55.9 | 61.8 | 68.3 |
| HRDA | 96.4 | 74.4 | 91.0 | 61.6 | 51.5 | 57.1 | 63.9 | 69.3 | 91.3 | 48.4 | 94.2 | 79.0 | 52.9 | 93.9 | 84.1 | 85.7 | 75.9 | 63.9 | 67.5 | 73.8 |
| MIC (HRDA) | 97.4 | 80.1 | 91.7 | 61.2 | 56.9 | 59.7 | **66.0** | 71.3 | 91.7 | 51.4 | **94.3** | 79.8 | 56.1 | 94.6 | 85.4 | **90.3** | 80.4 | 64.5 | 68.5 | 75.9 |
| LAMDA | **97.8** | **81.5** | **92.1** | **62.3** | **57.5** | **60.1** | 65.8 | **73.6** | **91.9** | **52.4** | 93.6 | **80.3** | **57.5** | **94.7** | **85.9** | 89.9 | **81.3** | **65.9** | **69.1** | **76.5** |
| **Synthetic-to-Real: Synthia→Cityscapes (Val.)** | | | | | | | | | | | | | | | | | | | | |
| ProDA | 87.8 | 45.7 | 84.6 | 37.1 | 0.6 | 44.0 | 54.6 | 37.0 | 88.1 | – | 84.4 | 74.2 | 24.3 | 88.2 | – | 51.1 | – | 40.5 | 45.6 | 55.5 |
| DAFormer | 84.5 | 40.7 | 88.4 | 41.5 | 6.5 | 50.0 | 55.0 | 54.6 | 86.0 | – | 89.8 | 73.2 | 48.2 | 87.2 | – | 53.2 | – | 53.9 | 61.7 | 60.9 |
| HRDA | 85.2 | 47.7 | 88.8 | 49.5 | 4.8 | 57.2 | 65.7 | 60.9 | 85.3 | – | 92.9 | 79.4 | 52.8 | 89.0 | – | 64.7 | – | 63.9 | 64.9 | 65.8 |
| MIC (HRDA) | 86.6 | **50.5** | 89.3 | 47.9 | 7.8 | 59.4 | **66.7** | 63.4 | **87.1** | – | 94.6 | **81.0** | 58.9 | 90.1 | – | 61.9 | – | **67.1** | 64.3 | 67.3 |
| LAMDA | **86.7** | 49.5 | **89.4** | 49.6 | 10.9 | 60.8 | 65.6 | 65.1 | 86.5 | - | 95.9 | 80.7 | 60.1 | 90.2 | - | 65.4 | - | 64.7 | 70.3 | 68.2 |
| **Day-to-Nighttime: Cityscapes→DarkZurich (Test)** | | | | | | | | | | | | | | | | | | | | |
| DANNet† | 90.0 | 54.0 | 74.8 | 41.0 | 21.1 | 25.0 | 26.8 | 30.2 | 72.0 | 26.2 | 84.0 | 47.0 | 33.9 | 68.2 | 19.0 | 0.3 | 66.4 | 38.3 | 23.6 | 44.3 |
| DAFormer | 93.5 | 65.5 | 73.3 | 39.4 | 19.2 | 53.3 | 44.1 | 44.0 | 59.5 | 34.5 | 66.6 | 53.4 | 52.7 | 82.1 | 52.7 | 9.5 | 89.3 | 50.5 | 38.5 | 53.8 |
| HRDA | 90.4 | 56.3 | 72.0 | 39.5 | 19.5 | 57.8 | **52.7** | 43.1 | 59.3 | 29.1 | 70.5 | 60.0 | 58.6 | 84.0 | 75.5 | 11.2 | 90.5 | 51.6 | 40.9 | 55.9 |
| MIC | 94.8 | 75.0 | 84.0 | **55.1** | 28.4 | 62.0 | 35.5 | 52.6 | **59.2** | 46.8 | 70.0 | **65.2** | 61.7 | 82.1 | 64.2 | 18.5 | **91.3** | 52.6 | 44.0 | 60.2 |
| LAMDA | **96.7** | **79.7** | **84.8** | 49.4 | **34.4** | **62.9** | 50.1 | 49.6 | 57.7 | 45.8 | **71.7** | 66.5 | 62.3 | 84.7 | - | - | 88.7 | **53.0** | 45.7 | **63.2** |
| **Clear-to-Adverse-Weather: Cityscapes→ACDC (Test)** | | | | | | | | | | | | | | | | | | | | |
| DANNet† | 84.3 | 54.2 | 77.6 | 38.0 | 30.0 | 18.9 | 41.6 | 35.2 | 71.3 | 39.4 | 86.6 | 48.7 | 29.2 | 76.2 | 41.6 | 43.0 | 58.6 | 32.6 | 43.9 | 50.0 |
| DAFormer | 58.4 | 51.3 | 84.0 | 42.7 | 35.1 | 50.7 | 30.0 | 57.0 | 74.8 | 52.8 | 51.3 | 58.3 | 32.6 | 82.7 | 58.3 | 54.9 | 82.4 | 44.1 | 50.7 | 55.4 |
| HRDA | 88.3 | 57.9 | 88.1 | 55.2 | 36.7 | 56.3 | 62.9 | 65.3 | 74.2 | 57.3 | 85.9 | 68.8 | 45.7 | 88.5 | 76.4 | 82.4 | 87.7 | 52.7 | 60.4 | 68.0 |
| MIC (HRDA) | 90.8 | **67.1** | **89.2** | **54.5** | **40.5** | 57.2 | 62.0 | **68.4** | 76.3 | **61.8** | **87.0** | **71.3** | 49.4 | 89.7 | 75.7 | 86.8 | 89.1 | 56.9 | 63.0 | 70.4 |
| LAMDA | **98.5** | 66.3 | 87.6 | 48.7 | 39.8 | **65.8** | **78.1** | 65.4 | **84.6** | 59.8 | 84.3 | 69.5 | **50.3** | **92.8** | **82.4** | **90.3** | **94.5** | **57.6** | **64.2** | **72.7** |
| **Multi-Source: GTA5 + Synthia → Cityscapes (Val.)** | | | | | | | | | | | | | | | | | | | | |
| MIC* | 97.61 | 81.34 | 92.01 | 62.38 | 56.91 | 60.23 | 67.00 | 72.26 | 92.70 | 51.88 | **95.11** | 81.22 | 57.21 | 95.00 | 87.10 | **88.87** | 80.62 | 65.32 | 68.02 | 76.46 |
| LAMDA | **98.17** | **82.73** | **92.57** | **63.48** | **57.37** | **61.03** | 66.17 | **74.56** | **92.89** | **52.78** | 94.48 | **81.62** | **58.50** | **95.08** | **87.15** | 88.12 | **81.58** | **66.72** | **69.09** | **77.06** |
| **Multi-Source: GTA5 + Synthia + BDD100K → Cityscapes (Val.)** | | | | | | | | | | | | | | | | | | | | |
| MIC* | 97.00 | 82.21 | 93.12 | 65.98 | **47.81** | **66.83** | 67.23 | **76.06** | 92.13 | 56.08 | 92.41 | 83.22 | 59.07 | **95.40** | 87.32 | **88.97** | 80.99 | 66.82 | 69.03 | 77.24 |
| LAMDA | **98.35** | **84.39** | **94.96** | **66.65** | 46.03 | 65.25 | **67.72** | 75.68 | **93.73** | **57.72** | **92.46** | **83.25** | **59.49** | 94.08 | **87.73** | 88.58 | **82.57** | **67.99** | **69.39** | **77.69** |
| **Source-Only: GTA5 → Cityscapes (Val.)** | | | | | | | | | | | | | | | | | | | | |
| MIC | **82.64** | **29.95** | 81.44 | 36.99 | 17.72 | 27.31 | 41.84 | 19.27 | **86.36** | 33.25 | **80.84** | 67.48 | 29.23 | 81.3 | 37.17 | 33.81 | 0.86 | 30.17 | 19.26 | 44.05 |
| LAMDA | 72.69 | 25.66 | **83.62** | **42.79** | **18.18** | **42.58** | **45.5** | **24.81** | 85.28 | **33.3** | 75.04 | **69.62** | **37.6** | **88.52** | **57.06** | **53.88** | **20.23** | **46.07** | **31.97** | **50.23** |
| **Source-Only: Synthia → Cityscapes (Val.)** | | | | | | | | | | | | | | | | | | | | |
| MIC | **86.71** | 41.16 | 81.28 | 10.42 | 0.17 | 39.69 | 43.41 | 29.12 | 82.68 | - | 76.51 | 67.41 | 28.45 | 81.16 | - | 46.77 | - | 42.59 | 38.51 | 50.06 |
| LAMDA | 85.74 | **43.32** | **84.56** | **11.66** | **1.06** | **44.77** | **51.92** | **51.91** | **85.35** | - | **80.94** | **72.24** | **40.78** | **85.11** | - | **47.61** | - | **53.77** | **60.08** | **56.30** |
| **Source-Only: Cityscapes → darkzurich (Val.)** | | | | | | | | | | | | | | | | | | | | |
| MIC | **95.32** | **76.33** | 78.3 | 38.72 | **46.47** | **48.43** | 17.13 | 31.81 | **35.44** | 21.62 | 0.8 | 25.66 | **24.95** | **70.71** | - | - | **44.54** | 15.97 | 30.57 | 36.99 |
| LAMDA | 94.79 | 75.79 | **78.84** | **40.53** | 41.84 | 44.2 | **53.21** | **34.81** | 35.23 | **33.1** | **1.54** | **33.43** | 9.32 | 67.05 | - | - | 43.9 | **16.05** | **38.24** | **39.05** |
| **Source-Only: Cityscapes → ACDC (Val.)** | | | | | | | | | | | | | | | | | | | | |
| MIC | 48.17 | 24.43 | **74.27** | **35.15** | 15.39 | 33.29 | 43.24 | 11.61 | **87.32** | 29.81 | 77.08 | 68.96 | **31.06** | 50.41 | 22.76 | 37.22 | 16.21 | 31.16 | 20.47 | 39.89 |
| LAMDA | **86.27** | **50.01** | 72.55 | 26.7 | **32.53** | **49.41** | **63.93** | **49.1** | 68.05 | **35.5** | **82.38** | 52.01 | 27.74 | **84.54** | **77.96** | **70.86** | **66.29** | **41.67** | **26.69** | **56.01** |

† Method uses additional daytime/clear-weather geographically-aligned reference images. * Method re-implemented by us since the original work did not report the results.

Table 1: The results of LAMDA for semantic segmentation with Close-Vocabulary.

alignment loss $\mathcal{L}^{align}$:

$$\mathcal{L} = \mathcal{L}^{\mathbb{S}} + \lambda^{\mathbb{T}}\mathcal{L}^{\mathbb{T}} + \mathcal{L}^{lmc} + \lambda^{align}\mathcal{L}^{align} \tag{8}$$

where $\lambda^{align}$ is the loss weight for $\mathcal{L}^{align}$. Note that there is also a pseudo label quality weight $\lambda^{lmc}$ in $\mathcal{L}^{lmc}$.

## 4 EXPERIMENTS

### 4.1 DATASETS SETTINGS

We give an overview about all the used dataset in the Appendix, including Cityscapes (Cordts et al. (2016)), GTA5 (Richter et al. (2016)), Synscapes (Wrenninge & Unger (2018)), SYNTHIA (Ros et al. (2016)), ACDC (Sakaridis et al. (2021)), Darkzurich (Sakaridis et al. (2020)), BDD100K (Yu et al. (2018)), KITTI (Geiger et al. (2012)) and Foggy Cityscapes (Sakaridis et al. (2018)).

### 4.2 IMPLEMENTATION DETAILS

For semantic segmentation UDA, we study synthetic-to-real (GTA5→Cityscapes, Synthia→Cityscapes), clear-to-adverse-weather (Cityscapes→ACDC), and day-to-nighttime (Cityscapes→DarkZurich) adaptation of street scenes. For object detection UDA, we study on Cityscapes→ Foggy Cityscapes. In the joint optimization, we follow DAFormer (Hoyer et al. (2022a)) and use MiT-B5 (Xie et al. (2021)) as encoder, and use AdamW with learning rate

| Method | Bus | Bcycl | Car | Mcycle | Person | Rider | Train | Trunk | mAP |
|---|---|---|---|---|---|---|---|---|---|
| **Close Vocabulary** | | | | | | | | | |
| DAFaster | 29.2 | 40.4 | 43.4 | 19.7 | 38.3 | 28.5 | 23.7 | 32.7 | 32.0 |
| SW-DA | 31.8 | 44.3 | 48.9 | 21.0 | 43.8 | 28.0 | 28.9 | 35.8 | 35.3 |
| SC-DA | 33.8 | 42.1 | 52.1 | 26.8 | 42.5 | 26.5 | 29.2 | 34.5 | 35.9 |
| MTOR | 38.6 | 35.6 | 44.0 | 28.3 | 30.6 | 41.4 | 40.6 | 21.9 | 35.1 |
| SIGMA | 50.4 | 40.6 | 60.3 | 31.7 | 44.0 | 43.9 | 51.5 | 31.6 | 44.2 |
| SADA | 50.3 | 45.4 | 62.1 | 32.4 | 48.5 | 52.6 | 31.5 | 29.5 | 44.0 |
| MIC(SADA) | 52.4 | 47.5 | 67.0 | 40.6 | **50.9** | 55.3 | 33.7 | 33.9 | 47.6 |
| LAMDA | **53.8** | **48.6** | **67.4** | **41.9** | 50.0 | **56.3** | **34.0** | **34.7** | **48.3** |

| Method | Seen | | | | | Unseen | | | mAP$_{seen}$ | mAP$_{unseen}$ |
|---|---|---|---|---|---|---|---|---|---|---|
| **Open Vocabulary** | | | | | | | | | | |
| | Bus | Bcycl | Car | Mcycle | Person | Rider | Train | Trunk | | |
| SADA | 37.2 | 43.1 | 61.7 | 30.5 | 42.9 | 18.4 | 2.1 | 12.5 | 43.1 | 11.0 |
| MIC(SADA) | 38.3 | 44.7 | 62.4 | 32.5 | 43.3 | 19.4 | 3.2 | 13.8 | 44.2 | 12.1 |
| LAMDA | **49.6** | **47.3** | **65.8** | **34.7** | **46.2** | **30.4** | **7.0** | **21.4** | **48.7** | **19.6** |

Table 2: The results of LAMDA for object detection with Close-Vocabulary and Open-Vocabulary.

| Method | Seen | | | | | | | | | | | | | | | Unseen | | | | | |
|---|---|---|---|---|---|---|---|---|---|---|---|---|---|---|---|---|---|---|---|---|---|
| | Road | Wall | Fence | Pole | TR.Light | Sky | Person | Rider | Car | Truck | Bus | Train | M.Cycle | Bicycle | mIoU | S.Walk | Build. | TR.Sign | Veget. | Terrain | mIoU |
| **Synthetic-to-Real: GTA→Cityscapes** | | | | | | | | | | | | | | | | | | | | | |
| DAFormer | 97.6 | 52.54 | 40.17 | 58.69 | 64.04 | 94.16 | 77.57 | 52.49 | 93.49 | 76.4 | 70.61 | 54.44 | 37.54 | 53.83 | 65.96 | 14.91 | 75.04 | 5.22 | 67.44 | 3.02 | 33.13 |
| HRDA | 96.24 | 47.67 | 42.08 | 59.74 | 62.35 | 93.23 | 77.09 | 49.62 | 93.74 | 72.42 | 72.19 | 38.3 | 61.89 | 66.53 | 66.65 | 25.24 | 74.41 | 4.53 | 72.13 | 5.54 | 36.37 |
| MIC | 97.44 | 55.31 | 46.23 | 59.23 | 66.23 | 93.68 | 77.39 | 52.32 | 94.23 | 81.61 | 75.69 | 44.51 | 57.52 | 63.48 | 68.92 | 18.09 | 76.17 | 10.17 | 75.99 | 3.17 | 36.72 |
| LAMDA | 97.72 | 54.67 | 44.98 | 60.06 | 63.72 | 93.72 | 77.60 | 50.99 | 94.29 | 81.74 | 77.46 | 64.74 | 61.96 | 63.11 | 70.48 | 27.17 | 75.01 | 10.82 | 76.51 | 7.79 | 39.46 |
| **Synthetic-to-Real: Synthia→Cityscapes** | | | | | | | | | | | | | | | | | | | | | |
| DAFormer | 83.43 | 50.86 | 22.34 | 53.8 | 61.55 | 83.81 | 79.27 | 48.73 | 90.16 | 23.12* | 63.95 | 0.0* | 40.15 | 68.81 | 54.99 | 13.86 | 52.73 | 2.06 | 20.61 | 2.92 | 18.44 |
| HRDA | 84.15 | 51.95 | 25.88 | 57.2 | 61.68 | 84.29 | 75.12 | 47.12 | 90.65 | 24.87* | 63.97 | 0.0* | 43.38 | 69.82 | 55.72 | 13.04 | 65.54 | 2.22 | 59.75 | 1.36 | 28.38 |
| MIC | 84.35 | 58.33 | 27.22 | 57.63 | 63.20 | 85.95 | 77.37 | 48.19 | 90.67 | 25.34* | 62.17 | 0.0* | 55.43 | 69.48 | 57.52 | 13.43 | 70.31 | 1.96 | 64.79 | 1.87 | 30.47 |
| LAMDA | 84.57 | 49.62 | 30.93 | 56.98 | 63.16 | 85.89 | 77.68 | 47.18 | 90.61 | 38.13* | 65.38 | 0.22* | 59.74 | 70.31 | 58.60 | 21.85 | 75.47 | 9.72 | 76.57 | 0.67 | 36.86 |
| **Day-to-Nighttime: Cityscapes→DarkZurich** | | | | | | | | | | | | | | | | | | | | | |
| DAFormer | 95.45 | 37.02 | 25.27 | 48.74 | 17.14 | 67.78 | 22.98 | 30.21 | 65.62 | - | - | 81.67 | 28.97 | 33.88 | 53.94 | 3.27 | 51.56 | 0.0 | 0.0 | 0.12 | 10.99 |
| HRDA | 95.3 | 42.98 | 45.67 | 43.65 | 10.6 | 71.97 | 26.5 | 17.69 | 70.19 | - | - | 84.81 | 16.8 | 30.46 | 54.12 | 0.0 | 52.64 | 0.0 | 0.05 | 3.66 | 11.27 |
| MIC | 95.58 | 36.94 | 22.45 | 47.3 | 16.26 | 65.78 | 36.95 | 40.38 | 72.06 | - | - | 84.74 | 19.94 | 29.5 | 55.21 | 13.93 | 69.41 | 0.09 | 1.71 | 0.0 | 17.03 |
| LAMDA | 95.32 | 38.72 | 46.47 | 48.43 | 17.13 | 70.32 | 25.66 | 24.95 | 70.71 | - | - | 84.54 | 19.97 | 30.57 | 55.69 | 4.82 | 76.29 | 0.0 | 23.0 | 0.0 | 20.82 |
| **Clear-to-Adverse-Weather: Cityscapes→ACDC** | | | | | | | | | | | | | | | | | | | | | |
| DAFormer | 89.13 | 37.39 | 32.27 | 56.57 | 69.85 | 82.71 | 60.59 | 37.48 | 85.67 | 76.67 | 87.52 | 86.9 | 48.51 | 40.35 | 63.69 | 6.67 | 58.22 | 6.50 | 43.27 | 0.0 | 22.93 |
| HRDA | 91.02 | 45.23 | 37.37 | 56.64 | 70.05 | 83.55 | 60.53 | 41.28 | 86.62 | 67.44 | 77.67 | 86.03 | 43.09 | 54.96 | 64.39 | 6.76 | 59.45 | 7.89 | 44.32 | 0.0 | 23.68 |
| MIC | 91.28 | 40.79 | 29.31 | 61.24 | 72.12 | 84.42 | 61.46 | 36.7 | 83.88 | 79.48 | 89.25 | 77.34 | 45.93 | 49.07 | 64.45 | 7.65 | 60.43 | 5.74 | 43.96 | 4.39 | 24.43 |
| LAMDA | 90.46 | 41.72 | 40.69 | 59.58 | 70.1 | 84.17 | 65.45 | 41.83 | 85.18 | 78.41 | 86.89 | 88.95 | 55.59 | 66.93 | | 9.17 | 63.92 | 9.80 | 45.17 | 0.35 | 25.68 |
| **Multi-Source: GTA5 + COCO→Cityscapes** | | | | | | | | | | | | | | | | | | | | | |
| DAFormer | 96.3 | 52.86 | 44.45 | 53.52 | 58.77 | 92.94 | 77.08 | 49.43 | 93.94 | 77.75 | 65.62 | 70.99 | 60.13 | 61.58 | 68.24 | 12.11 | 75.14 | 6.38 | 72.79 | 5.71 | 34.43 |
| HRDA | 97.29 | 53.21 | 44.0 | 56.87 | 64.25 | 93.67 | 77.6 | 52.54 | 94.07 | 80.46 | 74.36 | 75.98 | 63.35 | 66.36 | 70.93 | 23.87 | 74.86 | 10.20 | 73.76 | 4.59 | 37.47 |
| MIC | 97.38 | 53.23 | 46.81 | 56.52 | 66.38 | 93.85 | 77.82 | 52.84 | 94.05 | 82.44 | 70.7 | 80.43 | 64.62 | 65.94 | 71.61 | 24.25 | 76.27 | 16.58 | 77.02 | 3.82 | 39.59 |
| LAMDA | 97.35 | 53.65 | 46.03 | 55.25 | 66.72 | 92.46 | 77.25 | 53.49 | 94.08 | 83.73 | 75.58 | 81.51 | 63.99 | 65.39 | 71.95 | 42.04 | 79.43 | 18.69 | 79.77 | 5.38 | 45.06 |
| **Multi-Source: Cityscapes + COCO→ACDC** | | | | | | | | | | | | | | | | | | | | | |
| DAFormer | 89.15 | 48.56 | 38.0 | 58.24 | 65.13 | 82.07 | 60.45 | 44.52 | 87.12 | 68.94 | 70.55 | 86.4 | 42.16 | 52.49 | 63.84 | 16.28 | 64.56 | 7.88 | 56.34 | 0.0 | 29.01 |
| HRDA | 89.64 | 52.8 | 38.15 | 57.69 | 64.54 | 82.64 | 60.18 | 45.21 | 86.98 | 81.72 | 88.73 | 82.01 | 45.96 | 50.91 | 65.72 | 18.53 | 65.78 | 8.95 | 57.21 | 0.0 | 30.10 |
| MIC | 88.28 | 48.69 | 41.37 | 58.9 | 61.53 | 82.7 | 59.85 | 44.41 | 87.49 | 78.78 | 87.05 | 87.54 | 45.1 | 52.94 | 66.04 | 19.23 | 64.32 | 9.87 | 58.67 | 0.0 | 30.42 |
| LAMDA | 90.55 | 54.47 | 41.34 | 58.73 | 73.23 | 82.48 | 62.49 | 43.06 | 87.80 | 82.50 | 86.14 | 87.47 | 43.4 | 56.56 | 67.87 | 24.05 | 69.26 | 15.0 | 61.44 | 0.54 | 34.06 |

\* indicate that the result for this category was not labeled in the training dataset.

Table 3: The results of LAMDA for semantic segmentation with Open-Vocabulary.

$6 \times 10^{-5}$ for the encoder and $6 \times 10^{-4}$ for the decoder. The total training iteration is 60k. We set $\alpha = 0.9, \lambda^{\mathbb{T}} = 0.1, \lambda^{lmc} = 1, \lambda^{align} = 0.9$. Following previous works, we report the results in mIoU for segmentation, and mean Average Precision (mAP) with a 0.5 IoU threshold for detection.

### 4.3 MAIN RESULTS

We show that LAMDA is a universe UDA framework and achieves competetive performances to both multi-task joint-learning and single-task learning, under both source-only, single-source and multi-source UDA, with different language-driven settings. In total, we compare LAMDA with the following methods: ProDA (Zhang et al. (2021)), DAFormer (Hoyer et al. (2022a)), HRDA (Hoyer et al. (2022b)), MIC (Hoyer et al. (2023)), DAFaster (Chen et al. (2018)), SW-DA (Saito et al. (2019)), SC-DA (Zhu et al. (2019)), MTOR (Cai et al. (2019)), SIGMA (Li et al. (2022b)), SADA (Chen et al. (2021)).

#### 4.3.1 LAMDA WITH CLOSE-VOCABULARY

At the very first, we evaluate basic UDA performance on semantic segmentation and object detection, under various UDA scenes, without open-vocabulary setting.

**Single-Source:** As the most fundamental UDA scene, we firstly validate LAMDA on single-source UDA on semantic segmentation and object detection. At this setting, we unify the multi-task joint training with all the categories input in the language prompt,

| | LMC | HVA | Cross-modal Fusion | EMA | Segmentation (mIoU) | | | Detection (mAP) | | |
|---|---|---|---|---|---|---|---|---|---|---|
| | | | | | Close-Vocabulary | Open-Vocabulary | | Close-Vocabulary | Open-Vocabulary | |
| | | | | | | Seen | Unseen | | Seen | Unseen |
| Single-source | ✓ | | | | 68.28 | 63.43 | 34.39 | 37.68 | 45.76 | 13.35 |
| | ✓ | ✓ | | | 70.42 | 65.34 | 34.28 | 40.23 | 47.23 | 14.98 |
| | ✓ | ✓ | ✓ | | 71.32 | 66.95 | 35.37 | 42.54 | 48.65 | 15.34 |
| | ✓ | ✓ | ✓ | ✓ | **76.50** | **70.48** | **39.46** | **48.32** | **48.74** | **19.61** |

Table 4: Ablation study of components.

| Method | | Segmentation (mIoU) | | | Detection | | |
|---|---|---|---|---|---|---|---|
| | | Close Vocabulary | Open Vocabulary | | Close Vocabulary | Open Vocabulary | |
| | | | Seen | Unseen | | Seen | Unseen |
| Train Paradigm | Single-Task | 75.59 | 69.01 | 37.14 | 47.20 | 47.56 | 15.47 |
| | Multi-Task | 76.50 | 70.48 | 39.46 | 48.32 | 48.74 | 19.61 |
| Prompt | Ordinary | 73.91 | 68.32 | 35.99 | 45.78 | 46.02 | 16.19 |
| | Lean-able | 76.50 | 70.48 | 39.46 | 48.32 | 48.74 | 19.61 |

Table 5: Analysis of PLL and Prompt Engineering

**Source-Only:** The source-only UDA can better validate the generality of adaption to unknown target domain since the model can not access the target domain images.

**Multi-Source:** In the case of the labeled data available coming from multiple domains with different distributions, multi-source UDA is able to take advantage of the diverse and abundant knowledge to improve the generalization ability on target domain, while mitigating the mutual interference discrepancy between different sources. At this setting, in order to fit the multi-dataset training, we realize the mask decoding by multiplying text embedding and visual concept followed by the classification for segmentation, and we realize the region decoding with RPN head followed by the original contrastive alignment loss for detection.

The results of the above experiments settings are shown in Table 1 (segmentation) and Tabel 2 (detection). LAMDA obtains consistent improvements over all scenes and target domains.

### 4.3.2 LAMDA WITH OPEN-VOCABULARY

Further, we report the results of unified language-driven UDA for both semantic segmentation and object detection, with open-vocabulary setting. We mainly conduct experiments on single-source, and multi-source UDA.

**Single-Source:** We validate the generality of LAMDA on four datasets settings with three adaption scenes. Specifically, we select five categories from the common set of these target datasets as the unseen categories, and the rest others are as seen categories. During the training, the model can not access to the unseen categories and only be trained on the seen categories.

**Multi-Source:** We follow the same realization of decoders as full-language-driven experiments. And the setting of unseen categories are consistent with the single-source UDA settings.

The results of the above experiments are shown in Table 3. LAMDA also achieves remarkable improvements on both seen and unseen categories over all scenes and target domains.

### 4.4 IN-DEPTH ANALYSIS

**Ablation Study of Components:** We delve into each component of LAMDA. Specifically, we perform ablation study on LMC, HVA, Cross-model fusion, and EMA teacher. In the setting, "w/o LMC" indicates using randomly masking, "w/o HVA" indicates only aligning the source outputs with text embedding, "w/o EMA" indicates directly using the prediction of $f_\theta$ as LMC supervision. The results shown in Table 4 demonstrate their effectiveness.

**Closer Observation on PLL with UPD:** We show that the shared decoder with individual queries in UPD is a win-win for learning both detection and segmentation, since good box predictions are typically indicative of good masks, and vice versa. With UPD, PLL is able to effectively disentangle the parameter spaces to avoid the training conflicts in different tasks. In Table 5, we conduct multiple experiments on various PLL and UPD settings to prove the efficacy.

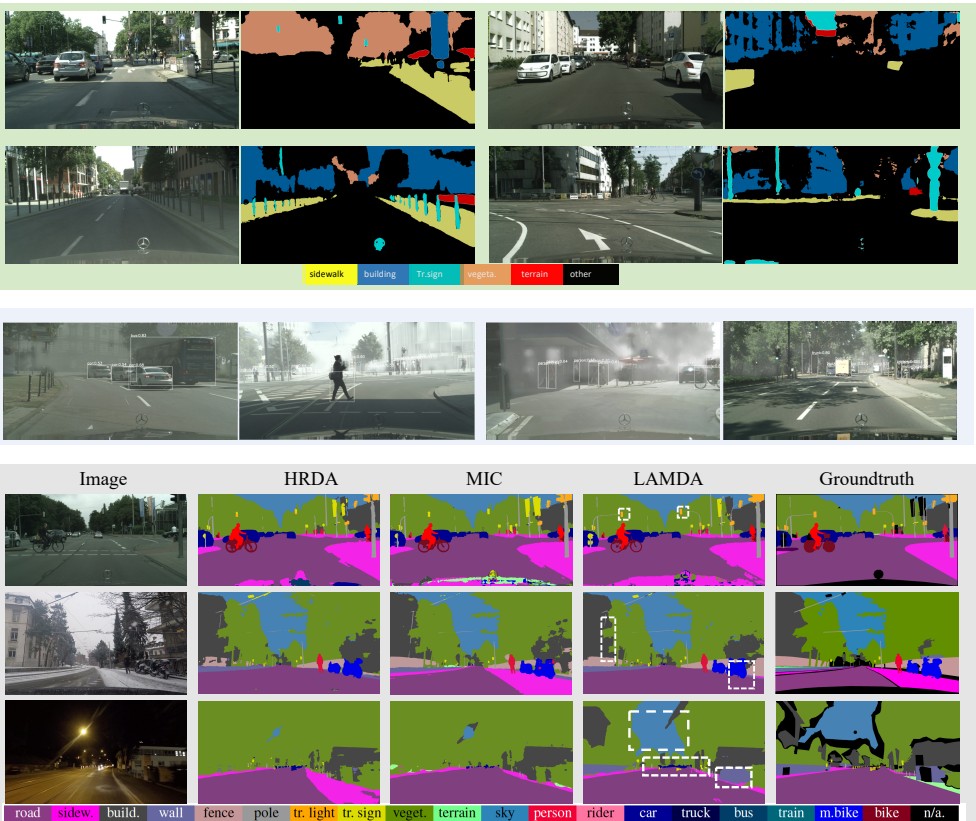

Figure 2: Upper: open-vocabulary segmentation ability of LAMDA on unseen target domain. Middle: open-vocabulary detection ability of LAMDA on unseen target domain. Lower: intuitive UDA improvements of LAMDA on target domain over previous methods (Synthia→Cityscapes, Cityscapes→ACDC, Cityscapes→DarkZurich).

**Prompt Engineering:** Further, we evaluation the impact of lean-able prompt templates on PLL by comparing with ordinary templates using the sentence "A photo of [CLASS/TASK]". As shown in Table 5, the learn-able prompts significantly outperform the ordinary ones regarding seen and unseen categories.

**Quantitative Results:** For more intuitive comparisons, we given some quantitative results in Figure 2. In the upper row, LAMDA demonstrates considerable open-vocabulary ability on unseen images with unseen categories in target domain. In the lower row, LAMDA shows intuitive improvements on target domain over previous methods including HRDA (Hoyer et al. (2022b)), MIC (Hoyer et al. (2023)), on both Synthia→Cityscapes, Cityscapes→ACDC, Cityscapes→DarkZurich. More Visualizations are in Appendix.

## 5 CONCLUSION

In this paper, we commence by conducting a comprehensive analysis of the prevalent limitations found in existing UDA methods. Then we introduce an innovative formulation of LAMDA which addresses the limitations and offers the unique advantage of unsupervised transfer to novel domains featuring custom vocabularies. We conduct extensive experiments on diverse datasets encompassing segmentation and detection tasks, which provides compelling evidence for the remarkable performance of LAMDA. Additionally, it is important to note that LAMDA serves as a universal framework with the potential to facilitate language-driven UDA in various other core vision tasks, including but not limited to instance segmentation and depth estimation.

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
