# APPENDIX OF *LAMDA: Unified Language-Driven Multi-Task Domain Adaption*

## 1  DATASETS DESCRIPTION

We given an intuitive overview of all the used datasets in our experiments in Table 1.

|  | Cityscapes | GTA5 | SYNTHIA | ACDC | Darkzurich | BDD100K | Foggy Cityscapes |
|---|---|---|---|---|---|---|---|
| Task | Seg & Det | Seg | Seg | Seg | Seg | Seg | Det |
| Image | 5000 | 24966 | 9400 | 1600 | 2567 | 9975 | 25000 |
| Scene | Real-world & Clear Weather & Day | Synthetic | Synthetic | Adverse Weather | Nighttime | Nature | Adverse Weather |

Table 1: The overview of all datasets.

## 2  MORE ABLATION STUDIES

**Settings of Language-guided Masking:** We show the influence of key settings in the Language-guided Masking, including patch_size and masking_ratio in Tabel 2. As shown that LAMDA achieves among the best performances in the range of patch_size between 64 to 128 and masking_ratio between 0.5 to 0.7.

## 3  MORE VISUALIZATIONS

### 3.1  OPEN-VOCABULARY ABILITY OF LAMDA

We show more visualization results in Figure 1.

### 3.2  COMPARISON ON UDA WITH STATE-OF-THE-ART METHODS

We show more visualization results in Figure 2.

## REFERENCES

| LMC Settings | | Mask Ratio | | | |
|---|---|---|---|---|---|
| | | 0.3 | 0.5 | 0.7 | 0.9 |
| Patch Size | 32 | 75.6 | 75.9 | 75.7 | 75.2 |
| | 64 | 76.0 | 76.3 | **76.5** | 75.9 |
| | 128 | 76.2 | **76.5** | 76.4 | 76.2 |
| | 256 | 75.2 | 75.0 | 74.9 | 75.8 |

Table 2: LMC Settings

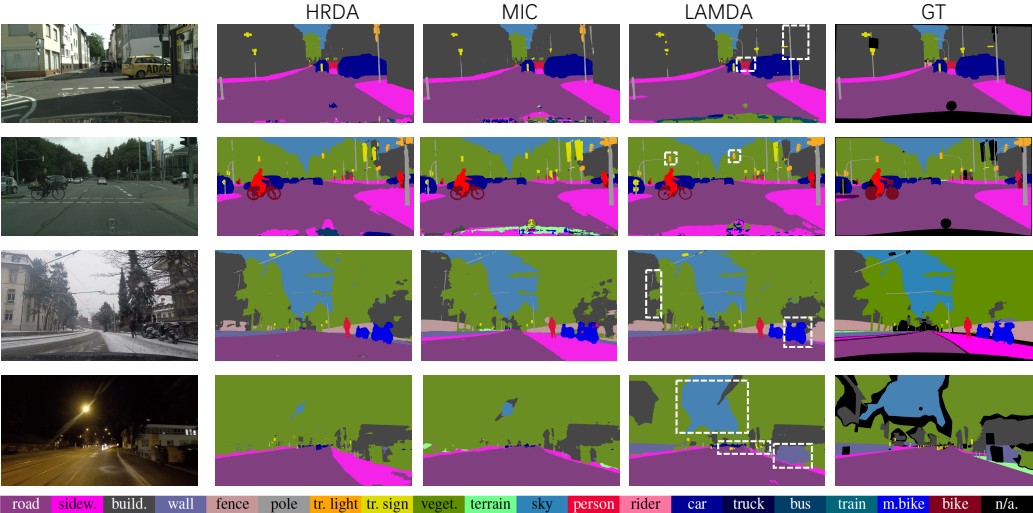

Figure 1: Comparison with state-of-the-arts: GTA5→Cityscapes (row 1), SYNTHIA→Cityscapes (row 2), Cityscapes→ACDC (row 3), and Cityscapes→DarkZurich(row 4).

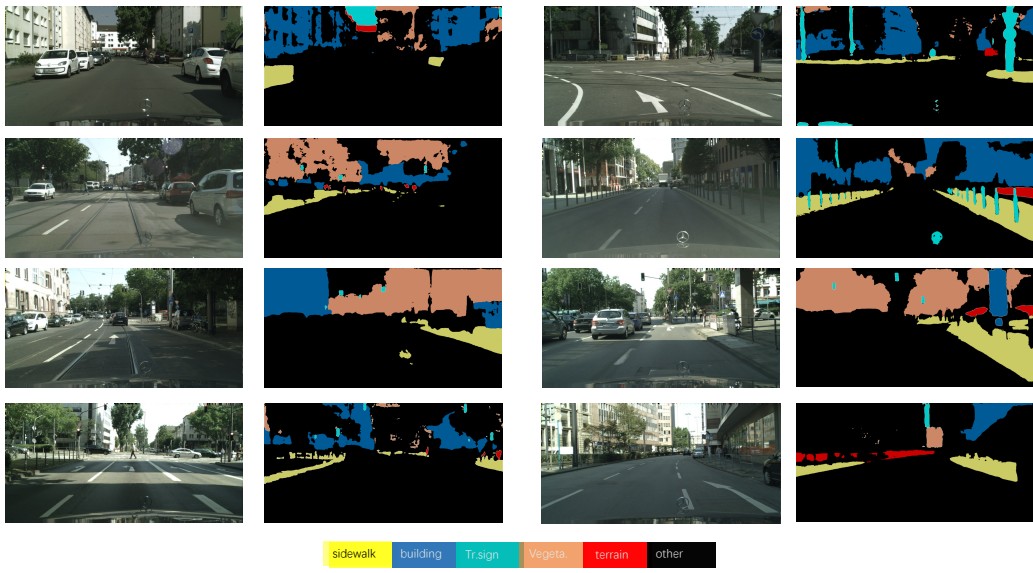

Figure 2: Open-vocabulary segmentation ability of LAMDA on unseen target domain.