# OpenReview forum: "LAMDA: Unified Language-Driven Multi-Task Domain Adaption"
_ICLR.cc/2024/Conference — ICLR 2024 Conference Withdrawn Submission_

### Official Review · Reviewer_VxaA · 2023-10-26

**Soundness:** 3 good
**Presentation:** 2 fair
**Contribution:** 3 good
**Rating:** 3
**Confidence:** 4

**Summary:**

This paper proposes a new method to incorporate vision-language backbones into the task of unsupervised domain adaptation, which would work universally for the tasks of semantic segmentation and object detection. They make three key contributions - which are universal backbones and proposal generation, language guided mask consistency and hierarchical visual-language alignment. They use Promptable Language-Task Learning (PLL) mechanism to collaboratively embed the language and task prompts. This is followed by mask-consistency on target samples with language-guided mask selection, and hierarchical VLA using proposals and prompts. Strong results are showcased on multiple detection and segmentation datasets on closed- and open-world settings.

**Strengths:**

- The paper tackles a very challenging problem of open-world domain adaptation in segmentation and detection which is a practical real world problem.

- The use of pre-trained CLIP models into UDA and open-world DA methods is interesting.

**Weaknesses:**

- I am not sure I understand where the open-world capabilities of the models come from. Detecting mask-proposals in an open-world has already been extensively explored in literature before (for example, Lseg[1] and FreeSeg[2]), so I do not think this paper proposes anything particularly targeted towards adapting to open-world categories.

- Adding to the above, the claimed contribution of joint architecture for segmentation and detection seems to be not fully true, as the architecture still has different decoding heads for segmentation and detection, different loss function, different task prompts and different training methodologies.

- Most importantly, the proposed components have very similar counterparts in existing literature. Specifically, the joint task-langauge prompt is very similar to the prompt design in FreeSeg [2], but FreeSeg was not even compared or mentioned. Likewise, the masked image consistency loss is already adopted to good effect in [3], but that work is not adequately cited. Finally, techniques similar to language-guided masking adopted in [4], which also needs to be highlighted.

- It is also not clearly stated if the text encoder is learnable or frozen? Also, if the text encoder is frozen, how are the prompts learned? The gradients are only passed to the prompts? This needs further explanation.

- Sec 3.2.1 states that a SWIN backbone transformer is adopted, while 4.2 states a DAFormer with MiT-B5 is adopted. This needs to be further clarified.

- A large portions of the method section, including the motivation and the design, is unclear from the current text. I would request the authors to make a more polished version in the next version of the paper.

- I am not sure I understand what is hierarchical in the hierarchical visual-langauge alignment? Isn't the loss very similar to the classic contrastive loss in CLIP, but at region level instead of image-level? Also, the sentence reads `In each training batch, the input images reflect three levels of domain hierarchy: intra-source, inter-source.`, what is the third?

- I would recommend the authors to explore the possibility of also including open-world segmentation methods (like LSeg or more) as possible baselines as they also train with CLIP and open-vocabulary capabilities.

[1]. Li, B., et al. "Language-driven semantic 427 segmentation." arXiv preprint arXiv:2201.03546 428 (2022).

[2] Qin, Jie, et al. "FreeSeg: Unified, Universal and Open-Vocabulary Image Segmentation." CVPR. 2023.

[3] Hoyer, Lukas, et al. "MIC: Masked image consistency for context-enhanced domain adaptation." CVPR. 2023.

[4] Li, Gang, et al. "Semmae: Semantic-guided masking for learning masked autoencoders." NeurIPS 2022.

**Questions:**

- Overall, the method seems to involve several distinct components which are not necessarily related to the broad goal of open-world domain adaptation. I would request the authors address the questions posed above, and I would be happy to raise my score.

---

> ### Author Response · Authors · 2023-11-15
> **Response to Reviewer VxaA**
>
> We are grateful for the recognition  regarding the challenging problem: open-world domain adaptation in segmentation and detection. We agree that this problem holds significant practical relevance in real-world scenarios and requires innovative solutions.
>
> Below are our clarifications regarding your concerns:
>
> 1. In our proposed method, we introduce the concept of universal open-world recognition in unsupervised domain adaptation (UDA), which endows the models with fine-grained and robust open-world recognition capabilities through the HVA. This represents a novel and practical task setting, and is one of our key contributions. While it is true that detecting mask-proposals in an open-world has been explored in existing literature, such as FreeSeg, our approach goes beyond conventional open-world recognition models that primarily focus on open-world scenarios within a single domain. We specifically address the challenges of multi-domain universal open-world recognition in transferred scenarios. This distinction sets our work apart and highlights the unique contribution we make in the field of open-world learning.
>
> 2. For the first time, we propose a unified multi-task open-vocabulary UDA framework. However, constructing a unified model in the visual domain is inherently more challenging than NLP, as there is currently no fixed established paradigm for visual tasks. We have made efforts:
>
>   ○ Decoding: mask decoding and region decoding are implemented within a unified decoder, which logically possesses different levels of output capabilities.
>
>   ○ Loss: We categorize the multi-grained task outputs into the most common forms of mask and box for open-world localization. The two losses essentially belong to the localization loss category and differ only in terms. For open-world recognition, we unify the computation  within a single process.
>
>   ○ Task Prompts: The role of task prompts is to mitigate the gap between tasks during training, while the network structure remains unified. Task prompts provide task-specific hints and are also usual in NLP.
>
>   ○ Training: Our framework adopts joint training, where categories of different task datas are encoded using a unified class prompt. Each batch contains multiple tasks, as the training process is highly unified.
>
> We hope our framework can  help the community make further progress towards  unified vision models.
>
> 3. The introduction and differences of relevant modules are as follows:
>
>   ○  joint task-langauge prompt: The concept of using a learnable joint task-langauge prompt does have some similarities to FreeSeg. However, unlike FreeSeg, our joint task-langauge prompt is also responsible for initializing different granularity queries for different tasks and utilized for self-training in cross-source to target domain adaptation.
>
>   ○ masked image consistency loss: We extensively compared the performance of MIC with our proposed LMC in Table 1 and 3, showing the significant improvements achieved by LMC compared to MIC in the context of language-guided masking.
>
>   ○ language-guided masking: Both SemMAE and our work aim to guide the masking process, yet we differ in specific guidance. We leverage rich language guidance in target domain to probabilistically mask objects. We will include related comparisons.
>
> 4. The text encoder is frozen. The category/task information and learnable prompt are fused together and input to the text encoder. Regarding gradients, the text encoder only passes gradients without updating parameters. Yet the learnable prompts still require parameter updates. Similar settings are in many related works, such as DenseClip[1].
>
> 5. In Sec. 3.2.1, we want to express image encoder can be any classical ViT backbones(e.g. Swin). Actually, we use MiT-B5  and will amend the ambiguity.
>
> 6. The core objective is to introduce the capability of open-vocabulary into a unified multi-task UDA framework. It aims to address the challenges of multi-task learning and transfer learning using a unified framework, by the modules: UPD, HVA, LMC, PLL.
>
> 7. We elaborate on the fours levels in HVA(See the Hierarchical Visual-Language Alignment part in the figure 1), which represent hierarchical visual-textual contrastive learning. As for loss, our idea is to align the outputs of different task levels by a contrastive learning in target domain. Theoretically, this provides more diverse positive and negative samples. Yet the contrastive loss in CLIP only focuses on single-level classification.
>
> 8. We want to address two challenges in UDA: 1) open-vocabulary transfer learning , 2)  unified framework for multi-task. In LSeg, direct model transfer yields inferior performance compared to DA operations. It also struggles to handle the unified execution of multi-grained tasks. So it is challenging to make a fair comparison between these approaches and our proposed DA optimization.
>
>
> [1]. "Denseclip: Language-guided dense prediction with context-aware prompting." CVPR 2022.

---

### Official Review · Reviewer_Nx7u · 2023-10-30

**Soundness:** 3 good
**Presentation:** 2 fair
**Contribution:** 3 good
**Rating:** 6
**Confidence:** 3

**Summary:**

This paper aims to propose an open-vocabulary domain adaptation framework based on vision-language models that can be applied to various core vision tasks, including object segmentation and semantic segmentation.
The proposed method is built by combining several technical components, but the core parts mainly contribute to unifying object detection and semantic segmentation tasks are: Unified Proposal Decoder (UPD) to decode the region to be decoded according to the task, and Promptable Language-Task Learning to explicitly embed task information into the text prompt.
Experimental results on datasets for domain adaptive semantic segmentation show that the proposed method can achieve accuracy comparable to or better than the state-of-the-art domain adaptation methods in both closed- and open-vocabulary settings.

**Strengths:**

The biggest contribution of this paper would be to address the challenging problem of constructing a unified framework for open-vocabulary object detection and semantic segmentation based on vision language models, a task that has not been well studied.

On the datasets used in the experiments, the proposed method achieves better accuracy than the state-of-the-art domain adaptation methods.

**Weaknesses:**

1. The goal of this paper is to provide a unified framework for various "core vision tasks". However, this paper focuses only on object detection and semantic segmentation, and it is not clear how well it will perform on other tasks.


2. In reality, the proposed method uses specialized modules and technical elements (region decoding and mask decoding) for object detection and semantic segmentation, respectively. Due to these parts customized for each downstream task, it may be a bit questionable whether we can say that the proposed method is truly a unified framework.


3. Despite the considerable complexity of the overall design of the proposed method, the results of the ablation study in Table 4 show that EMA, a well-known idea in various learning tasks, provides the largest gain. This seems to obscure the technical innovation brought by this paper.


4. One of the main components of the proposed method, Language-Guided Masked Consistency (LMC), generates masks (pseudo-labels) based on the output of the teacher model to (ideally) always mask out object regions. Unlike MLM/MIM/MIC, which randomly masks regions of the image, the proposed LMC always requires that objects be predicted from surrounding non-object regions, which seems to have the potential to enhance undesirable context bias (in an extreme example, if a cow is in the driveway, it would be recognized as a car), is there any justification or discussion on this point?


5. Several details of the proposed method are missing. Specifically, I could not find details on "Task-Conditioned Initialization" depicted in Fig. 1 and "Unified Proposal Decoder (UPD)".


6. I would say more related papers on open-vocabulary object detection/segmentation could be included in the paper. For example, [a] introduces a head architecture that conditions bounding box regression and mask segmentation by text embedding vectors for open-vocabulary instance segmentation. Inclusion of these highly relevant papers will clarify the technical contribution of the proposed method.

[a] Tao Wang, Learning to Detect and Segment for Open Vocabulary Object Detection, in CVPR 2023.


7. Some minor errors.
* Fig. 1: Lean-able -> Learnable?


* Fig. 1: The curly arrow from "Target Domain Masked Image x_1^T" to "Target Domain Masked Image x_2^T" should be from "Target Domain Image x_2^T" to "Target Domain Masked Image x_2^T".


* I could not find "Unified Proposal Decoder (UPG)" in Fig. 1, which should be explicitly depicted for improving readability.


* The abbreviation MLM first appears in Sec. 3.2.2, but what it stands for is never explained. While it is clearly masked language modeling from the context but should be specified


* "three levels of domain hierarchy: intra-source, inter-source." Considering Fig. 1, I am guessing this should be "four levels of domain hierarchy: intra-source, multi-source-target, and multi-source-masked-target."

**Questions:**

My questions are listed in descending order of importance in the Weaknesses section above. I would be grateful if the authors could answer as many of my questions as possible.

---

> ### Author Response · Authors · 2023-11-15
> **Response to Reviewer Nx7u**
>
> We appreciate the recognition of our work in addressing the challenging problem of constructing a unified open-vocabulary UDA framework for detection and segmentation. As highlighted by the reviewers, this task has not been extensively studied, and we are delighted that our proposed method makes a significant contribution in this area.
>
> Below are our clarifications regarding your concerns:
>
> 1. Existing works in the field of UDA predominantly concentrate on detection and segmentation tasks. Therefore, our work aligns with the current research direction in the domain. In the future, if there are other DA visual understanding tasks, our framework is designed to support task extensions.  Other visual tasks such as depth estimation and grounding can also be represented within our framework using masks and bounding boxes respectively.
>
> 2. The region and mask decoding in our proposed method are performed within a unified decoder. which logically possesses different levels of output capabilities. This hierarchical output can cover the majority of fine-grained visual understanding task outputs. It is worth noting that there is currently no standardized paradigm for a unified model in the field of computer vision. Achieving a unified framework in vision is more challenging than NLP. While there are related works such as OpenSeed[1], the key difference lies in our focus on hierarchical understanding tasks within a unified framework for open-vocabulary UDA.  We aim to explore and contribute to the direction of unified visual models.
>
> 3. EMA, in conjunction with LMC, is a method used to implement self-training in UDA. Its objective is to guide learning in the target domain with pseudo teacher supervision, which effectively leverages the distillation capability of LMC. Without the incorporation of EMA, the full potential of LMC remains untapped. By introducing self-training, LMC's distillation ability can be maximized. We position EMA as the last experiment in the ablation study, which may have caused ambiguity in assessing its effectiveness within that specific ablation setting. In the field of domain adaptation, self-training is a widely recognized and effective approach for leveraging pseudo-labels to guide learning in the target domain.
>
> 4.  The objective of LMC is to enhance the effectiveness of masking through contextual guidance, rather than randomly masking regions in original MIM/MIC. During training, we employ a high probability of masking object regions but not always to encourage the model to learn from more challenging scenarios, thereby improving its generalization capability. By appropriately increasing the masking probability, we can reduce redundant information to a greater extent, which is beneficial for domain transfer in the end. Regarding extreme cases, we have conducted relevant tests where masking all object regions resulted in a decrease in performance. However, through adaptive learning within LMC and its integration with the overall network training optimization, LMC actively avoids masking all object regions to the best of its ability.
>
> 5. Apologies for the confusion, we provide detailed explanations as follows:
>
>   ○ Task-Conditioned Initialization: In Fig. 1, Task-Conditioned Initialization refers to the fusion of learnable task prompt initialization parameters with visual information and then stacked with the parameters of region query and mask query. In essence, this initialization incorporates task information, enhancing the effectiveness of proposal outputs during training for different tasks, as compared to the original nn.parameter initialization (in PyTorch).
>
>   ○ Unified Proposal Decoder (UPD): In the main network module in Figure 1, Region Decoding and Mask Decoding together form the UPD module.  The Proposal Decoder unify the generating of different outputs such as masks or regions based on context information from the input image and queries.
>
> 6. In the paradigm of open-vocabulary recognition, both the paper [2] and our proposed method employ a proposal objects and textual semantics alignment approach to enable open-vocabulary recognition. We will provide a citation to [2]  in our manuscript to clarify the technical contribution and establish the relevance of our approach. Thank you for bringing this paper to our attention, and we appreciate your suggestion to include more related papers in the field of open-vocabulary object detection/segmentation.
>
> 7. Thank you for pointing out the minor errors in the manuscript, which will be rectified in the final version.  Specifically, for "intra-source, inter-source, source-target, multi-source-masked-target," our intention is to enhance the ability of open vocabulary recognition by using a hierarchical positive and negative sampling approach.
>
> [1]."A simple framework for open-vocabulary segmentation and detection." CVPR 2023.
>
> [2]. "Learning to detect and segment for open vocabulary object detection." CVPR 2023.

---

### Official Review · Reviewer_QBkk · 2023-11-05

**Soundness:** 2 fair
**Presentation:** 2 fair
**Contribution:** 2 fair
**Rating:** 5
**Confidence:** 3

**Summary:**

The method proposed in this paper addresses the vocabulary-specific gap across different tasks and datasets, achieving commendable results in unsupervised domain adaptation benchmarks.

**Strengths:**

This paper advances the performance of unsupervised domain adaptation benchmarks by introducing and employing more advanced  techniques.

**Weaknesses:**

1. The paper contains an many modules and abbreviations, making it challenging to read and comprehend.
2. The image encoder is not adequately introduced, and it is unclear whether it was pre-trained on Im1k or 21k, along with its scale for comparison.
3. Some of the techniques in the paper lack references. For instance, in Section 3.2, the paper mentions the use of MIM & EMA without referencing relevant literature. Similarly, the Unified Proposal Decoder in Section 3.2.1 appears to be derived from Detr and related variants. Sections 3.2.2 and 3.2.3 share a similar issue.
4. This paper leverages advanced pre-trained models like CLIP and incorporates several advanced designs to outperform the baseline, which is not very surpursing.
5. The paper mentions using the CLIP text encoder to address vocabulary-specific issues, which seems more akin to tackling open-vocabulary problems. What are the distinctions between the two, and how does the proposed approach compare to open-vocabulary methods?

**Questions:**

See weakness.

---

> ### Author Response · Authors · 2023-11-15
> **Response to Reviewer QBkk**
>
> We would like to express our sincere gratitude to the reviewer for recognizing the advancements made in our work regarding open-vocabulary UDA benchmarks. We have strived to introduce and employ more advanced techniques to enhance the performance of domain adaptation, and we are pleased to see that our efforts have been acknowledged.
>
>
> Below are our clarifications regarding your concerns:
>
>
> 1. We apologize for the excessive use of abbreviations in our paper, which may have made it challenging to read and comprehend. We will make sure to provide detailed explanations for each abbreviation in order to facilitate a better understanding of our model. The following abbreviations mentioned in the question are specific to our paper:
>
>   ○ LAMDA: Unified Language-Driven Multi-Task Domain Adaptation. This represents the method and concept we propose.
>
>   ○ LMC: Language-Guided Masked Consistency. It utilizes language guidance to determine the masked probabilities for patches in the target domain, leading to improved domain adaptation.
>
>   ○ HVA: Hierarchical Visual-Language Alignment, focuses on aligning textual and visual information across images and domains.
>
>   ○ UPD: Unified Proposal Decoder, is a module in our framework designed to unify the tasks of object detection and segmentation. It generates masks for segmentation or bounding boxes for object detection.
>
>   ○ PLL: Promptable Language-Task Learning, is a training method we propose to enhance the learning of multiple tasks, such as detection and segmentation.
>
>
> 2. In the LAMDA framework, the image encoder can be replaced with various classical ViT backbones. In our paper, we utilized the MIT-B5 model, which was pre-trained on ImageNet-1K dataset. We apologize for the lack of clarity.
>
> 3. We apologize for the lack of accurate referencing in our paper, which may have affected the reading experience. Regarding the techniques mentioned in Section 3.2,
>
>   ○ MIM is a learning method, and one classic approach is [1]
>
>   ○ EMA is  specifically in the form of self-training for UDA, which involves using a teacher network to generate pseudo-labels in the target domain. A relevant reference is: [2]
>
>   ○ The Unified Proposal Decoder employ a unified decoder to generates universal masks and bounding boxes for segmentation or detection tasks. The query-based paradigm is originally inspired by the DETR model, but we formulate the universal architecture.
>
>   ○ In Section 3.2.2, the LMC module utilizes language guidance to determine the masked probabilities in the target domain, improving domain adaptation. It is our original contribution.
>
>   ○ In Section 3.2.3, the HVA module aligns textual and visual information across batches, domains.
> We will ensure to provide proper and complete references in the final version of the paper.
>
> 4. CLIP and similar models, trained on large amounts of data, demonstrate good transferability and generalization. However, they have limitations when it comes to specific tasks. For instance, in the detection and segmentation tasks addressed in our paper, CLIP only possesses image-level understanding, whereas detection and segmentation require region-level and pixel-level comprehension, respectively. Direct transfer is insufficient to solve these tasks. The core contribution of our paper lies in addressing the open vocabulary recognition issue in unified task domain adaptation using VLM models. This entails solving the following key challenges: 1) designing a unified multi-task model (LAMDA/UPD), 2) leveraging VLM to achieve open vocabulary recognition and enhance performance in the target domain (HVA), 3) addressing the lack of attention mechanism in traditional self-training methods during domain adaptation (LMC), and 4) handling the differences between multi-task learning in a unified model (PLL).  We would like to emphasize that our work goes beyond the utilization of advanced pre-trained models and incorporates novel designs to improve performance in unified task domain adaptation.
>
>
> 5. The core objectives of our paper are to address two key challenges: 1) achieving open-vocabulary recognition capability in the target domain during domain adaptation, and 2) resolving the need for training in  multi-task domain adaptation scenarios. We employ a visual-text alignment approach, our distinction lies in enhancing the model's performance in the target domain, which requires not only addressing open-vocabulary recognition but also effectively utilizing the target domain's specific features. Our proposed approach differs from traditional open-vocabulary methods by emphasizing the improvement of performance in the target domain, while also considering the implementation of open-vocabulary recognition.
>
> [1]  "Masked autoencoders are scalable vision learners." CVPR 2022.
>
> [2]  "Mean teachers are better role models: Weight-averaged consistency targets improve semi-supervised deep learning results." NeurIPS 2017

---

### Author Response · Authors · 2023-11-15
**Response to all the Reviewers**

We would like to express our heartfelt gratitude to all the reviewers for their time and dedication. We sincerely appreciate the recognition of our work, and are grateful for the chance to address the concerns raised by the reviewers.

We would like to emphasize that our paper is among the first to propose a novel task setting by introducing open-vocabulary recognition capability in the context of Unsupervised Domain Adaptation (UDA) for detection and segmentation tasks. This sets it apart from conventional open-vocabulary recognition models, as our focus is primarily on the open-vocabulary issue in the UDA setting, which distinguishes it from conventional works such as LSeg and FreeSeg. Recently, we have come across related works [1] that combine UDA with open-vocabulary classification tasks. However, our task represents the first attempt to integrate UDA with more fine-grained open-vocabulary detection and segmentation tasks.


[1]. Huang, Jiaxing, et al. "Prompt Ensemble Self-training for Open-Vocabulary Domain Adaptation." arXiv preprint arXiv:2306.16658 (2023).